# Whole Genome Amplification in Preimplantation Genetic Testing in the Era of Massively Parallel Sequencing

**DOI:** 10.3390/ijms23094819

**Published:** 2022-04-27

**Authors:** Ludmila Volozonoka, Anna Miskova, Linda Gailite

**Affiliations:** 1Scientific Laboratory of Molecular Genetics, Riga Stradins University, LV-1007 Riga, Latvia; linda.gailite@rsu.lv; 2Department of Obstetrics and Gynaecology, Riga Stradins University, LV-1007 Riga, Latvia; anna.miskova@rsu.lv

**Keywords:** whole genome amplification, preimplantation genetic testing, embryo, next generation sequencing, massively parallel sequencing, haplotype, multiple displacement amplification, degenerate oligonucleotide primer

## Abstract

Successful whole genome amplification (WGA) is a cornerstone of contemporary preimplantation genetic testing (PGT). Choosing the most suitable WGA technique for PGT can be particularly challenging because each WGA technique performs differently in combination with different downstream processing and detection methods. The aim of this review is to provide insight into the performance and drawbacks of DOP-PCR, MDA and MALBAC, as well as the hybrid WGA techniques most widely used in PGT. As the field of PGT is moving towards a wide adaptation of comprehensive massively parallel sequencing (MPS)-based approaches, we especially focus our review on MPS parameters and detection opportunities of WGA-amplified material, i.e., mappability of reads, uniformity of coverage and its influence on copy number variation analysis, and genomic coverage and its influence on single nucleotide variation calling. The ability of MDA-based WGA solutions to better cover the targeted genome and the ability of PCR-based solutions to provide better uniformity of coverage are highlighted. While numerous comprehensive PGT solutions exploiting different WGA types and adjusted bioinformatic pipelines to detect copy number and single nucleotide changes are available, the ones exploiting MDA appear more advantageous. The opportunity to fully analyse the targeted genome is influenced by the MPS parameters themselves rather than the solely chosen WGA.

## 1. Introduction

Applications of whole genome amplification (WGA) are widespread in clinical and scientific practices dealing with a limited amount of starting genetic material that is often of poor quality. Such practices include human identification in forensic and archeological investigations, microbiological studies, and prenatal and preimplantation genetic testing (PGT). PGT is clinically performed to exclude chromosomal or genetic pathology in the embryo with the aim of increasing embryo implantation rates or conceiving a healthy child in a family with a certain monogenic disease. Embryo genetic testing for monogenic disorders could only be realized after the pioneering of polymerase chain reaction (PCR) in 1985, and was first performed in 1989 [1]. With time, PGT has undergone significant methodological and approach changes in terms of clinical indications, embryological toolkit (manual versus laser-assisted biopsy and freezing versus vitrification), analysed material (blastomeres, polar bodies, trophectoderm cells and even morula cells), and, most importantly, downstream genetic applications, to improve conventional PCR (nested PCR, multiplex PCR, and fluorescent PCR).

In 1992, two well-known WGA technologies were developed–primer extension preamplification PCR (PEP-PCR) and degenerate oligonucleotide primer PCR (DOP-PCR) [2,3]. WGA involves the use of random or partially random primers with the aim of efficiently replicating limited amounts of a target genome in a sequence-independent manner. Due to the nature of the procedure, any WGA is prone to drawbacks such as incomplete genomic coverage and amplification bias. Notwithstanding these shortcomings, WGA advancement was a turning point for PGT. Access to an almost endless supply of otherwise minute amounts of amplified DNA of a single or a few cells allowed for multilocus analysis, leading to a more versatile and safer PGT practice.

The majority of PGT cycles are performed to exclude the main cause of failed embryo implantation–chromosomal aneuploidies [4]—termed PGT-A. The next most common reason to perform PGT is structural chromosomal rearrangements in one of the parents’ karyotypes, most often Robertsonian or reciprocal translocations. Classical PGT for structural rearrangements, termed PGT-SR, can only distinguish between balanced and unbalanced embryos, without an option to identify carrier embryos. Being straightforward and not requiring any specific preparation, prerequisites of PGT-A and classical PGT-SR are successful embryo biopsy and WGA. This is because aneuploidy or, in a broader concept, copy number variation (CNV) screening is performed in a genome-wide manner commonly using array comparative genomic hybridization (aCGH) or massively parallel sequencing (MPS), thus requiring substantial amounts of genetic material.

In contrast, embryo assessment for a particular single gene or monogenic disorder (PGT-M) can be particularly challenging and prone to several biological and technical drawbacks. These include crossover events adjacent to the locus of interest, consanguinity, de novo mutations (biological), decreased methodological resolution, DNA contamination, and allelic dropout (ADO), which is amplification failure affecting one or both of the parental alleles (technical) [5]. Classical PGT-M requires customization in advance of the clinical test in order to construct family haplotypes—a practice that is time consuming and only useful for the particular gene/locus or sometimes only for the family being tested. Haplotyping is essential to track the transmission of a disease allele to an embryo and is commonly done through the assessment of polymorphic markers—short tandem repeats (STRs) or single nucleotide polymorphisms/variations (SNPs/SNVs)—adjacent to the mutation locus [6,7]. Thus, being an inherently multifactor analysis, PGT-M also heavily relies on the merits of WGA.

As the first step of PGT-M and PGT-SR cycles frequently involves WGA, this allows for additional aneuploidy screening in the same biopsy sample—a practice termed multifactor PGT. The exclusion of chromosomal aneuploidies in PGT-M and PGT-SR cycles has been reported to be associated with improved pregnancy rates [5,8]. As a consequence, it is being increasingly requested in clinics. However, running several methods in parallel complicates and prolongs the practical application of mainstream PGT. To overcome the obstacles associated with multifactor PGT performed in a parallel manner, application specialists are pursuing time-saving and simplified comprehensive PGT (cPGT) approaches, combining PGT-M, PGT-A, and PGT-SR in a single assay (Figure 1) and universally suiting every case. In such a scenario, PGT-M is performed based on a genome-wide haplotyping approach requiring minimal or no custom preparation. Being clinically attractive, cPGT puts serious demands on the technical and molecular execution of the universal application and especially on WGA, the starting point of any embryo genetic testing in the era of genomics.

## 2. Requirements of WGA in PGT

Unlike any other application, PGT deals with very precious and sensitive material, namely the DNA of a developing embryo. Thus, any actions and manipulations of an embryo or its parts are associated with certain limitations and peculiarities. First, biopsy of an embryo needs to be done in the least traumatizing manner, with successful removal of as few cells as possible during the first attempt. The safest practice is now accepted to be trophectoderm biopsy of a day-5 developing embryo [9,10]. Briefly, biopsied cells are washed, tubed, and transferred to the genetic testing unit. Thereafter, the blastocyst can be cultivated until day 6 for the 24-h PGT protocol or vitrified depending on the downstream manipulations and/or clinical indications.

In addition to the high demands placed on embryological personnel, they are also placed on the WGA performance itself. The most important requisite originally required from WGA for PGT—to provide an abundance of template DNA—is still as important today. One diploid cell contains just 6–7 ng of DNA, whereas WGA is expected to yield at least a few micrograms of the reaction product to suffice downstream PGT applications. Furthermore, more important than the amplification yield per se are several characteristics of the obtained WGA product: (i) good coverage of the genome, i.e., covering as much as possible of all 3 × 10^9^ nucleotides comprising the human genome; (ii) uniformity of the amplified material, i.e., regions of the genome need to be amplified homogeneously to allow the reliable quantification of CNVs; (iii) preserved original nucleotide sequence to enable reliable genotyping, i.e., avoiding loss of one or both gene copies without introducing artificial sequence variation [11].

Thus, for PGT, we have WGA that is inherently ADO and representation bias prone, and deals with precious and sensitive material—minute amounts of embryonic DNA. The greatest price for unsuccessful WGA is rebiopsy or in the worst-case scenario a lost embryo due to failure to genotype. Hence, in order to amplify the whole genome in a sequence-independent manner, several WGA requirements need to be proposed: (i) use of enzymes with high processivity and high fidelity, i.e., with proofreading activity ensuring lower error rates as well as longer amplicons; (ii) increasing the number of priming events by using universal primers and adjusting annealing/reannealing conditions (i.e., cycling conditions); (iii) reducing the complexity of the genome prior to amplification, e.g., by fragmentation.

Since there is currently no WGA deemed to be the gold standard in the ever developing and changing field of PGT, choosing one can be difficult. The aim of this review is to provide a comprehensive insight into the role and application of WGA in PGT, mainly focusing on cPGT solutions employing MPS. In the following sections, we detail the types of WGA commonly utilized in PGT, provide a comparison of WGA-associated drawbacks hampering embryo genome analysis across the different WGA techniques, and review cPGT solutions exploiting different WGA types.

## 3. WGA Types

### 3.1. PEP-PCR

Zhang et al. first described the application of WGA from a single cell in 1992. They termed this method primer extension preamplification PCR. Their new technique exploited random 15-base oligonucleotides non-specifically binding to a target genome. Theoretically, the primer was composed of a mixture of 1 × 10^9^ different sequences and was estimated to capture and amplify at least 78% of the haploid genome of a single sperm cell as assessed through targeted loci analysis [2]. Initially, the PEP-PCR protocol was claimed to be too lengthy and no better than direct single-cell PCR by other groups [12]. However, following modifications which resulted in a better genome recovery, this protocol has been successfully applied to various cells (amniocytes, chorionic villi, blastomeres) and has enabled several genetic loci of interest in human genetic diseases to be examined with a good amplification efficiency [13,14,15]. The development of this WGA technique was a breakthrough in the field of human PGT, as it allowed for the first time simultaneous multiple locus analysis with the further opportunity to validate the findings. Although contemporary applications do not (or very rarely) employ the original PEP-PCR approach, its principle has been integrated into all successive WGA developments.

### 3.2. DOP-PCR

Primer extension preamplification was quickly followed by the development of the more widely adopted DOP-PCR protocol, first described by Telenius et al. [16], to complement the cytogenetic analysis of flow-sorted chromosomes. Degenerate oligonucleotide primer PCR uses partially degenerate primers binding to many sites of the targeted genome. Variations of DOP-PCR primers include oligos with six degenerated bases in the middle, flanked by defined sequences at both ends (traditional DOP-PCR primer 5′-CCGACTCGAGNNNNNNATGTGG-3′) or oligos with a random 3′ end and partially fixed 5′ sequence–oligos with increased degeneracy, also termed tagged random primers. In all cases, due to the primers’ properties, DOP-PCR synthesis occurs in two stages (Figure 2). First, the degenerate part of the primer binds to the target genome over several low-stringency annealing cycles (~25–30 °C), followed by specific primer binding at higher temperatures, allowing for an exponential increase in the amplified material (second stage high-stringency cycles) [11,17]. Employing a proofreading enzyme, e.g., *Pwo*, and increasing the annealing and extension time improve the performance of the assay, resulting in fewer erroneous sequences, longer PCR fragments and increased genomic coverage [18,19]. DOP-PCR was first used in human PGT by Wells and colleagues to increase the amount of cytogenetic information for the comparative genomic hybridization (CGH) of single blastomeres [20]. A decade later, they used the same approach for trophectoderm biopsies [21].

Being PCR-based methods, both PEP and DOP PCRs generate relatively short products no longer than a few kilobases, preferentially amplifying a few hundred-long fragments [22].

### 3.3. MDA

The next important technical advancement in the application of WGA methods was the development of multiple displacement amplification (MDA). Originally termed multiply-primed rolling circle amplification [23], MDA was initially developed to amplify circular templates, but was subsequently modified for the amplification of linear ones. MDA exploits the unique properties of bacteriophage φ29 DNA polymerase (phi29) [24]. This enzyme possesses proofreading activity and has the capacity to perform strand displacement DNA synthesis for more than 70,000 nucleotides under isothermal conditions without dissociating from the template. Thus, MDA is considered a non-PCR-based technique. In MDA, phi29 extends random hexamer primers, producing multibranched structures (Figure 3). Improvement in yield is achieved by using 3′-end thiophosphate-modified random hexamer primers that are exonuclease resistant. The DNA fragments produced a range from a few kb to hundreds of kb in length [22,25].

The first successful clinical application of MDA in human PGT was reported by Handyside and colleagues. A combination of direct mutation loci testing of *CFTR* and linked markers was carried out on cleavage stage embryos, with the ADO rate reaching 16% and an amplification failure of 8% when amplifying single lymphocytes and 0% when amplifying blastomeres [26]. The simplicity of MDA usage over other techniques has also been acknowledged by other groups [27]. Furthermore, Handyside and colleagues subsequently described Karyomapping—a hallmark technology in PGT-M based on genome-wide haplotyping that also exploits MDA for embryo WGA [28].

### 3.4. MALBAC

Significantly later, in 2012, Zong et al. reported the development of a new WGA technology termed multiple annealing and looping-based amplification cycles (MALBAC). MALBAC introduces quasi-linear preamplification to reduce the bias associated with non-linear amplification (first stage PCR). The preamplification is initiated with a pool of random primers, each having a common 5′ 27-nucleotide sequence and 3′ eight random nucleotides, hybridizing to the template DNA at low temperatures (15–20 °C, capable of hybridizing at 0 °C) [29]. Thus, this stage resembles the MDA principle, with the difference that the MALBAC preamplification stage consists of multiple (5 to 12) annealing-extension-denaturation-looping steps, rather than isothermal synthesis [11]. Following hybridization, using an elevated temperature (~65 °C), a DNA polymerase with strand displacement activity is used to generate semi-amplicons of variable lengths. Eventually, amplification of these semi-amplicons produces full amplicons with complementary ends, which allow for full amplicon looping and prevent them from further amplification and cross-hybridizations (Figure 4). Thus, the key to MALBAC is not to make copies of copies, but rather to only make copies of the original genomic DNA by protecting the amplification products. The preamplification stage is then followed by a regular exponential amplification of the full amplicons, where oligos complementary to the common 27-nucleotide sequence are used (second round amplification), generating enough DNA for downstream applications [25]. The amplicon length of the final product can range between 0.2 and 2.0 kb [30].

The potential application of MALBAC in human PGT was first demonstrated by sequencing both polar bodies and oocyte pronuclei to phase the genomes of donors and determine the crossover maps of their oocytes, as well as assess for aneuploidy and SNVs in disease-associated alleles [31]. A year later, MALBAC was validated for 24-chromosome aneuploidy screening of cleavage stage embryos [32].

### 3.5. Hybrid WGA Methods

As the various WGA technologies were being established, hybrid WGA techniques combining features of PCR-based and MDA-based WGA were also becoming commercially available. The most well-known hybrid WGA methodology is PicoPLEX, originally introduced by Rubicon Genomics and now marketed as SurePlex by Illumina and a few other companies. While information on the exact principle of these commercially available kits is not easily available due to patents and continuous upgrades, they all utilize two-stage PCR. During the first preamplification stage, template DNA is amplified utilizing either: (i) non-self-complementary/self-inert primers which preclude primer dimer formation [11]; or (ii) primers allowing for the looping of full amplicons, similar to the ones used in MALBAC [33,34,35]. In both cases, amplicons produced during the first preamplification step do not participate further in the preamplification reaction (i.e., do not produce copies of the copies). Primers used in the preamplification reaction are composed of two sections—a degenerated sequence at the 3′-end responsible for frequent priming in the genome and a fixed sequence at the 5′-end, resembling DOP-PCR primers. The second stage PCR amplifies the molecules synthesized during the first stage reaction [11]. PicoPLEX was launched in 2009 and is still widely used. PGT performed on trophectoderm biopsies using PicoPLEX in combination with BlueGnome’s BAC arrays (later Illumina, now discontinued) and subsequently MPS began a major revolution in the availability of human PGT and forever changed the face of clinical in vitro fertilization [36,37,38]. MALBAC and PicoPLEX commercial kits amalgamating WGA and next-generation sequencing (NGS) library construction into a single procedure are also available, enabling the laborious NGS library preparation protocol to be shortened. All the commercial WGA kits mentioned in this review as originally described by the authors of the original publications are summarized in Appendix A.

Figure 5 illustrates the timeline of the development of the different WGA types as well as the associated hallmark achievements of PGT.

## 4. WGA Drawbacks

The following section describes the most common WGA drawbacks and their nature. A more detailed comparison of these phenomena across the different WGA types is provided later in the review.

### 4.1. ADO and Incomplete Genomic Coverage

It is thought that, once the template DNA concentration falls below a certain input level, the probability of obtaining a complete template genome, especially with the expectation of uniform amplification, decreases dramatically. At very low initial DNA concentrations, random and difficult to predict (stochastic) effects dictate whether a particular genomic region will be amplified or not [42]. The so-called ‘Monte Carlo effect’ states that “the lower the abundance of any template, the less likely its true abundance will be reflected in the amplified product” [43]. Incomplete genomic coverage is apparent from such events as allelic or locus dropout—one of the major drawbacks of WGA in human PGT. The phenomenon of ADO is widely recognized—perceived from the first PGT trials performed using direct PCR—and is the main reason for adopting haplotyping—the simultaneous amplification of causative genes together with linked polymorphic markers—as the gold standard in PGT [5]. It is important to note that ADO can arise not only from incomplete genomic coverage, but also from preferential amplification of one of the alleles. Therefore, ADO is a complex phenomenon not only influenced by the molecular technique, but also inherent to single-cell applications in general.

Targeted amplification of WGA-amplified material demonstrated that ADO rates almost double in comparison to direct PCR without WGA [8]. Thus, the cruciality of detecting ADO in the backdrop of widespread WGA utilization has not lost any of its importance [5,8]. Having said that, comparing ADO rates of different WGA techniques using targeted loci analysis may not reveal the overall picture of what is actually occurring. In such cases, ADO will also depend on the target primer design and particular DNA locus analysed, not to mention the different PCR conditions and chemistry used. However, Bonnette et al. claimed that multiplex STR kits used in human identification, e.g., Thermo Fisher’s AmpFLSTR, are ideal for evaluating the genome-wide performance of WGA techniques [18] and demonstrated that STR peaks on electropherograms do not show any stutter peaks after DOP-PCR WGA. However, in contrast to commercial kits, custom-designed STR analysis can be hampered by multiple stutter peaks, especially when using a nested PCR approach [44,45].

### 4.2. Amplification Bias

Amplification bias, also termed PCR drift, results when certain regions/amplicons within a multitemplate reaction are preferentially amplified relative to the entire pool of potential templates [29]. As expected, the lower the concentration of the initial template, the more prominent the effects of amplification bias. Amplicon representation bias is very much affected by primer composition, i.e., degree of degeneracy of the primer. Substantial overamplification can result due to complementarity between the 3′ region of the primer and the genomic sequence [29,42]. Furthermore, WGA primer binding and synthesis efficiency depend on characteristics inherent to the input DNA itself, e.g., GC content and spatial/secondary DNA conformation. Similar to PCR-based WGA approaches, isothermal MDA’s principle of hyperbranching also negatively affects sequence-dependent bias and causes overamplification in certain genomic regions and under-amplification in others [25].

### 4.3. Chimera and Non-Template Amplicons

Apart from representation bias, WGA can produce a certain degree of chimera amplicons—a kind of artificial amplicon mapping to different parts of the genome that are not physically linked. The dominant type of chimeras are intra-chromosomal translocations, suggesting that chimeras are produced by neighbouring amplicons randomly connecting on the same chromosome [46]. While the formation of chimeric PCR fragments has been attributed to MDA [47] and further demonstrated for other WGA methods [25,47], no issues arising from this phenomenon have been reported in PGT. Of note, no significant preference has been recorded in the distributions of chimeras and hotspots among chromosomes; however, preferences in overlap length and GC content have been shown to be pertinent to the sequence denaturation temperature, highlighting a direction of action for reducing chimeras [48]. Non-template amplicons are associated with contamination and are common to any amplification employing random or degenerate oligos [42]. They can be addressed by implementing good laboratory practices.

### 4.4. WGA-Independent Improvements

There are a few measures that can be undertaken to increase the efficiency of any WGA technique. The foremost one is to perform a trophectoderm biopsy. Irrespective of the widely demonstrated clinical benefits [49,50], performing a blastocyst stage biopsy (which implies biopsy of several trophectoderm cells) also affords a number of technical advantages compared to a single-cell biopsy with subsequent WGA. First, in comparison to a single cell, several cells increase the number of template genomes, thus smoothing the abrogating WGA effects described directly above. For example, Handyside and colleagues investigated the relationship between ADO and input cell number. They demonstrated that ADOs occurred randomly at a frequency of approximately 16% in single-cell amplifications, but were undetectable when the number of input cells was increased to 10–20 [26]. This finding was subsequently confirmed by Rechitsky et al. [8]. Additionally, Tzvetkov et al. reported that concordance calls between MDA-amplified and unamplified DNA increased significantly when 20 ng of input DNA was used instead of 6 ng [51].

Moreover, Dimitriadou et al. demonstrated that the quality of single-cell aCGH data, especially to detect segmental chromosomal imbalances, is dependent on the cycle of the analysed cell. Cells undergoing DNA synthesis (S phase) show a more fluctuating aCGH profile than cells in the G0/G1 phase. Again, a blastocyst stage biopsy addresses this issue, with the simultaneous analysis of several cells ameliorating cell cycle discrepancies, leading to a better and more uniform amplification and ultimately a more confident genetic diagnosis [52]. Thus, trophectoderm biopsy is the logical approach to maximize the efficiency of the different WGA techniques and, consequently, for successful human PGT as a whole.

The phenomena of amplification bias and incomplete genomic coverage can be further driven by manipulations such as freezing and the prolonged storage of biopsy material prior to WGA, which cause a certain degree of template DNA degradation. Even slight discrepancies in amplification efficiency can result in a substantial divergence in the proportional representation of amplicons in the final product, even after relatively few cycles [53]. Therefore, performing WGA immediately after procuring the biopsy without freezing or storing of the biopsy material before WGA minimizes any form of degradation and increases amplification efficiency. Furthermore, the European Society of Human Reproduction and Embryology (ESHRE), for good practice guidelines on PGT-M, recommend subjecting 30–50 single-cell or few-cell samples to WGA and downstream applications to demonstrate a methodology’s successful performance ahead of clinical application, as well as inclusion of negative and positive controls to monitor the WGA reaction [6].

## 5. Performance Comparison of Different WGA Types

The main parameters for which preimplantation embryos are assessed during PGT are CNV and SNV. It is important to assess not only whole chromosome aneuploidies but also structural CNVs. The assessment of SNVs in PGT can be performed for haplotyping purposes or for the analysis of mutation loci themselves. Reliable analysis of these genetic variations directly depends on the DNA polymerase, chemistry, and cycling conditions of the WGA used. Another important aspect to consider when comparing/demonstrating the performance of different WGA assays is cell type. Ideally, for PGT, performance comparison needs to be conducted using DNA from embryonic cells of the same developmental stage, as WGA efficiency can vary depending on the cell type assessed.

As mentioned earlier, amplification bias is the main parameter influencing CNV assessment, while genomic coverage evenness is the main parameter influencing SNV analysis. Of note, the major drawbacks inherent to single-cell analysis and the WGA procedure itself are further influenced by the MPS conditions, e.g., library preparation PCR can further affect the uniformity of genomic coverage. Increasing the number of reads and, consequently, the sequencing depth of MPS, increases genomic coverage as well, and vice versa. Thus, when analysing/comparing the performance of different WGA types, close attention must also be paid to conditions unrelated to the WGA techniques themselves, i.e., the ensuing downstream applications.

### 5.1. Mappability of Reads

As mentioned earlier, WGA produces a certain fraction of non-specific amplicons. Similarly, using MPS, there will always be a percentage of reads not mapping to the reference genome. The source of these unmappable reads is junk reads arising from the formation of primer dimers, short DNA fragments, and non-target genomes [25,54]. For example, when comparing the WGA types MDA, MALBAC, and PicoPLEX using 5× sequencing depth, it was demonstrated that the fraction of unmapped reads for all the amplifiers was low (0.035% mean fraction of unmapped reads) [54]. In accordance, similar numbers were reported by Deleye and colleagues, who compared four different WGA types (REPLI-g, Ampli1, DOPlify and Picoseq; please refer to Appendix A listing all the commercially available WGA kits). Using a lymphoblastoid cell line and 0.3× sequencing depth, the mappability of reads ranged from 97.0 ± 3.1% to 99.7 ± 0.01% [55]. In addition, at 0.1× sequencing depth, Hou et al. obtained 98.36% of mapped reads for MDA, 97.68% for MALBAC, and 89.31% for DOP-PCR; there was no significant difference in the GC content of unmapped reads among the methods [46].

In contrast, a much higher unmappable read percentage was demonstrated using deeper sequencing (30×). Specifically, the percentage of unmappable reads for GenomePlex (Sigma Aldrich, St. Louis, MO, USA) and PicoPLEX (Rubicon Genomics, Ann Arbor, MI, USA) (which are considered the same chemistry WGA kits) reached 64 and 66%, respectively. REPLI-g MDA kit (Qiagen) and MALBAC (Yikon Genomics, Shanghai, China) showed 18% and 22% of unmapped reads, respectively [25]. Another group detailed 35–40% of unmappable reads from SurePlex- and MALBAC-amplified single white blood cells (sequencing depth unknown) [30]. Such large percentages of unmappable reads were proposed to be due to the presence of universal adapter reads, which are WGA-independent, as well as primer sequences and substantial contributions from small amplicons [25].

These sequencing data clearly show that the same WGA can yield different proportions of mappable reads under different sequencing depths. By increasing the sequencing depth, the proportion of mappable reads decreases.

Another important parameter influencing the effective usage of sequencing reads is duplicate reads. De Bourcy et al. examined several different WGA types and found that, on average, 7–45% of mapped read pairs from a high sequencing depth run were PCR or optical duplicates [54]. More specifically, different MDA kits produced 1.73–6.52% of duplicates, while DOP-PCR duplicate reads reached 39.24% [46]. The uniform/even distribution of duplicate reads negatively affects the sequencing efficiency, reducing the amount of effective coverage. However, if the distribution of duplicate reads is uneven, this leads to the more important problem of reduced uniformity of coverage, as discussed below.

Investigations have demonstrated that reaction volume, reaction yield, and even the number of hands-on steps (perhaps increasing the likelihood of contamination) influence the proportion of mappable reads [54]. Thus, it is important to distinguish genuine WGA-driven unmappable reads from WGA-independent sequences. Such a big data difference demonstrates the reduced reproducibility of the performances of various WGA assays in combination with different downstream applications using different cells. When cost effectiveness is considered, a low unmappable read rate is desirable for high effective coverage. The ESHRE PGT-SR/PGT-A Working Group recommends that, of the total number of reads, 70–80% should align to the genome, with lower percentages indicating contamination, DNA degradation, or suboptimal WGA [56].

### 5.2. Uniformity of Coverage and CNV Analysis

As discussed earlier, deficient uniformity of coverage is intrinsic to single-cell applications and is further affected by WGA. In PGT-A, it is essential to have uniform coverage or to have bioinformatic algorithms to manage PCR bias.

In 2002, it was reported that DOP-PCR leads to a strong amplification bias, with individual loci differing in copy number by four to six orders of magnitude [57]. As MDA-based amplification is isothermal, as opposed to PCR-based WGA methods, one common assumption has been that the MDA technique is immune to GC bias. However, DNA regions with a high localized GC content also prove to be problematic for isothermal amplification, leading to reports of under-representation caused by reduced DNA polymerase processivity and poor DNA priming in high GC areas [42]. Furthermore, it has been shown that the amplification bias in MDA progressively worsens with greater fold amplification, whereas MALBAC and PicoPLEX appear relatively insensitive to reaction gain [54].

It has subsequently been demonstrated that DOP-PCR and other PCR-based WGA methods exhibit reasonable uniform amplification with reduced regional amplification bias and outperform MDA in terms of CNV detection using arrays or NGS [30,54]. Despite a high duplication ratio and limited genome recovery, Hou et al. found that DOP-PCR still displayed the highest accuracy for CNV detection (≥1 Mb), with a mean sensitivity and specificity of 94.15% [46]. It has also been demonstrated that DOP-PCR provides the flattest CNV raw data without normalization, while non-PCR-based MDA creates variations along the genome that are not reproducible and cannot be smoothed via normalization. Furthermore, MALBAC’s sequence-dependent bias is reproducible and gives the flattest CNV after normalization [25]. Li and colleagues showed that the WGA of both SurePlex and MALBAC can reliably and accurately detect known disease CNVs in the range of 3–15 Mb at the correct genomic breakpoints [30]. Generally, it is acknowledged that all types of WGA provide a distribution of reads uniform enough to be able to accurately call CNVs [25,46,55].

### 5.3. Genomic Coverage and SNV Calling

In the era of arrays, genome representation assessment directly depended on array resolution, which, when compared to MPS, could never cover the full genome or exome. Thus, array genome representation percentages cannot be compared to the ones derived from NGS data. Using 10 K SNP arrays, MDA was estimated to amplify 99.82% of the genome [58]. Using MPS with ~25× mean sequencing depth, MDA covered 72% of the genome and MALBAC achieved up to 93% genomic coverage of single cancer cells [29]. MDA using either phi29 or *Bst* DNA polymerase has been widely reported to achieve a high physical coverage (>90%) from a single-cell genome or exome at a high sequencing depth (typically >30× or at least 15× average sequencing depth) [46,59,60]. In contrast, GenomePlex and PicoPLEX (kits with the same chemistry) covered only 39% and 52% of the reference genome, respectively (30× sequencing depth) [25]. Reduced genomic coverage has also been acknowledged for DOP-PCR [46,59].

Conversely, shallow sequencing depth runs retrieve only very limited fractions of the genome. For example, MDA attained 8.84% genomic coverage at a mean sequencing depth of ~0.5×, which was slightly higher than that of MALBAC (8.06%) [46].

Taking account of the percentages of genomes retrieved at deep and shallow sequencing depths, it is evident that the WGA methods themselves are capable of amplifying significant proportions of the target genome, and it is in fact the selected parameters of the downstream applications (e.g., MPS) that limit the detection of the amplified genome.

Consequently, areas not covered by the DNA polymerase during WGA—or importantly not captured during downstream molecular applications—are expected to occur as dropouts. The ability to successfully amplify and detect SNVs/SNPs is particularly important for PGT-M, as well as for PGT-SR, to distinguish between balanced carrier and non-carrier embryos. SNV detection efficiency/dropout rates directly depend on genomic coverage and therefore show similar tendencies to those observed for genomic coverage across the different WGA types. Indeed, MDA-based technologies perform better in terms of SNV detection. For instance, using long-fragment sequencing at a depth of 80×, Peters et al. demonstrated that MDA allowed for the detection of up to 93% of heterozygous SNVs [61]. Similar data have subsequently been presented for short-read sequencing [46]. In the same year as Peters et al.’s report, Zong and colleagues documented better SNV detection efficiency for MALBAC WGA, which is an MDA-based assay. Specifically, using a single cancer cell, MALBAC enabled 76% of SNVs to be detected, whereas it was only 41% for MDA [29]. Additionally, the advantage of employing MDA over PicoPLEX in terms of SNV detection was illustrated using both arrays and NGS by the group that developed Haplarithmisis [62]. Furthermore, Huang et al. compared five different commercially available WGA kits and determined the lowest ADO rate for MALBAC (21%) and the highest ADO rate for GenomePlex (76%) [25]. It has also been established that the low genomic coverage of DOP-PCR makes it an inappropriate approach for performing genotyping at a base pair resolution [59].

### 5.4. False Positive Rate

No less important than undetectable alleles are false alleles that occur due to errors made by the DNA polymerase of the WGA or downstream application assay. Usage of high-affinity (not easily dissociated from the DNA strand), robust (working through tough reaction conditions, e.g., without a purification step) and high-fidelity (i.e., maintaining Watson-Crick base pairing) DNA polymerases, in addition to possessing 3′ > 5′ proofreading activity, is the key to reducing the number of false positives during any PCR/DNA synthesis.

MDA exploits phi29 DNA polymerase which has 3′ > 5′ proofreading activity, though is not thermostable. In Sanger sequencing experiments sampling 500,000 base pairs, the false positive rate of phi29 was found to be relatively low (9.5 × 10^−6^) and not statistically different from that of paired WGA-unamplified samples [63]. MALBAC’s first step utilizes *Bst* DNA polymerase, which is similar to phi29 but lacks proofreading activity [64]. The false positive rate due to errors generated in the first MALBAC cycle and propagated in the later amplification has been reported to be ~4 × 10^−5^ [25,29], slightly higher than that of phi29. De Bourcy and colleagues, exploiting molecular barcoding, which allows true variants to be discriminated from sequencing errors, demonstrated that the per-base error rate for MDA was at least one order of magnitude lower than the rates of MALBAC and PicoPLEX rates. It was concluded that the use of a DNA polymerase with proofreading activity (i.e., phi29) is a major advantage of MDA for analysing SNVs [54]. In addition to comparing the ADO rates of five different commercially available WGA kits (see above), Huang et al. also compared their error rates and found them to be fairly similar, ranging from 9.6 × 10^−4^ for DOP-PCR to 8.2 × 10^−5^ for MDA [25]. It can be concluded that, due to their incomparably lower rates, false positives are of much less concern than ADOs.

## 6. Comprehensive PGT Solutions Utilizing Different WGA Protocols

### 6.1. The Beginning of the Massively Parallel Sequencing Era in Human PGT

The first reports of the use of MPS in PGT-A were published in 2013 [31,37,38]. Everything started with the low-depth, low-coverage whole genome sequencing (WGS) to detect CNVs when no high genome coverage was needed. In turn, there was a need to develop bioinformatic pipelines specifically addressing WGA-induced bias, as it might limit the sensitivity and specificity of CNV detection. Zhang et al. addressed this need by creating a bioinformatic pipeline centred on GC correction for the removal of DOP-PCR WGA-induced bias in low-coverage sequencing data (4–9.5% of the genome covered). They established the so-called dynamic threshold/frame approach to access CNVs more accurately—one of the principles still applied today for CNV analysis in PGT, which also does not require a control sample(s) [38]. Next, Hou and colleagues sequenced the genome of MALBAC-amplified polar bodies and pronuclei of human oocytes, demonstrating that even ~1× average genome depth yields reproducible CNV calls with ~1 Mb of resolution [31]. Soon after Hou et al.’s study, it was confirmed that a 0.04–0.07× average genome depth and 3–5% of genomic coverage can serve as the standard for PGT-A analysis using MPS [32,37].

In the early days of MPS, it became clear that it provides a better signal-to-noise ratio and resolution than aCGH, simply due to advances in technology. NGS specificity for aneuploidy calling was demonstrated to be 99.98% with a sensitivity of 100% [65]. Exceptional multiplexing opportunities and falling NGS costs facilitated the smooth transition of PGT-A towards monumental MPS exploitation. At the beginning of the MPS era in PGT, the majority of applications used SurePlex WGA, as MPS data were often validated by the formerly-used established aCGH protocols that widely employed SurePlex WGA (e.g., [65,66]).

Since the adoption of MPS for PGT-A, the designs of new applications have been moving towards better coverage, higher resolution, and more reliable data in general that can be used not only for CNVs, but also for cPGT including whole-genome haplotyping and direct mutation loci assessment. There are numerous turnkey solutions in PGT exploiting all the described WGA types, including hybrid ones. This section details MPS-exploiting, up-to-date solutions aspired to a universal/cPGT. While microchips are still in use, only the foremost ones are described here (i.e., Karyomapping and Haplarithmisis).

### 6.2. Karyomapping and Haplarithmisis

Karyomapping was one of the first applications to exploit SNP arrays in PGT. At the time, it was a rapid alternative to the targeted STR typing approach used as standard in PGT-M. Genome-wide SNV haplotyping allows Karyomapping to detect CNVs, meiotic trisomy, monosomy, triploidy, parthenogenetic activation, uniparental heterodisomy, as well as patterns of genomic duplication seen in, for example, hydatidiform moles—all in a single workflow. The assay requires single- or multi-cell embryo biopsy amplified by an isothermal MDA as the starting material [28,67].

Haplarithmisis—an extension of Karyomapping—is similarly a genome-wide generic PGT tool that originally exploited SNP arrays and MDA for single-/few-cell WGA. A computational pipeline evaluates the observed versus expected SNP probe’s intensity values for each allele in the sample, thus allowing detection of CNVs, the mitotic or meiotic nature of chromosomal anomalies (with the exception of monosomies), low-grade mosaicism, as well as proper ploidy (e.g., enables the distinction of aberrant tetraploid from aberrant diploid) [62].

As with any other genome-wide haplotyping technology, Karyomapping and Haplarithmisis are not universally applicable to all PGT-M cases, such as those where the analysis is not available for both parents, or cases with a highly consanguineous background, or ones with de novo mutations. Indeed, Rechitsky et al.’s analysis of 2780 couples revealed that Karyomapping was not suitable for as many as one-third of families referred for PGT-M [8]. Karyomapping and Haplarithmisis are now applicable for both SNP arrays and MPS platforms.

### 6.3. OnePGT

A commercial NGS-based solution that integrates PGT-A, PGT-SR, and PGT-M in a single workflow—OnePGT by Agilent Technologies—has recently been released to the market. The protocol exploits MDA for the embryo WGA and reduced representation WGS and has been verified on a few Illumina-sequencing platforms. To deduce haplotype inheritance, the embedded PGT-M pipeline utilizes principles of Haplarithmisis and was developed by the same group. Both the PGT-A and PGT-SR pipelines are based upon read-count analysis in order to assess CNVs. Inherent to haplotyping methodologies, the processing of additional family members such as the proband or a grandparent is required for haplotype establishment [68].

### 6.4. MARSALA

The MPS application termed Mutated Allele Revealed by Sequencing with Aneuploidy and Linkage Analyses (MARSALA) combines low-coverage genome sequencing for PGT-A and the targeted enrichment of mutation loci and linked SNVs for PGT-M. A prerequisite for MARSALA is the genome sequences of the parents. In the absence of an affected relative of the parents, an affected embryo identified by direct calling of the causative SNV or embryo haplotyping can be used as an equivalent of the proband for linkage analyses [69,70]. The application uses MALBAC for embryo WGA, an aliquot of which is reamplified with a pair of target-specific primers, and then the targeted PCR products are mixed with the native WGA for NGS. In this way, the existence of the point mutation and aneuploidy can be detected in one NGS run. The region of interest can be sequenced to ultra-high coverage (>1000×), still maintaining an accurate CNV measurement throughout the whole genome. It has been demonstrated that, in contrast to MDA, using MALBAC for single-cell WGA linkage analyses can be achieved with only 2× sequencing depth [69,70].

Mai et al. recently described a similar target sequence enrichment/semi-targeted approach. Briefly, after the low-stringency cycles of the DOP-PCR protocol, the WGA reaction is supplemented with a pool of mutation loci and SNV-specific primers. High-stringency PCR conditions enable the amplification of both the target sequences and the whole genome fragments created during the low-stringency PCR cycles. While not necessary, to ensure the sufficient breadth and depth of coverage for mutation and adjacent SNV loci, a secondary target enrichment PCR can be performed using the initial WGA product. The target PCR product is then spiked back into the WGA, typically at 1:10 to 1:20 concentrations [71].

### 6.5. MaReCs

Technologically similar to MARSALA and developed by the same group, Mapping Allele with Resolved Carrier Status (MaReCs) is a PGT-SR methodology. Whilst MaReCs does not require pre-clinical work-up to phase haplotypes, it is performed in two stages. First, embryo CNVs are analysed by a high-coverage, high-resolution WGS. Secondly, targeted NGS analysis is performed for 60 adjacent SNVs flanking the translocation breakpoint in a manner similar to MARSALA to perform haplotyping [72]. The availability of at least one chromosomally imbalanced embryo (so-called reference embryo) is essential for the pipeline to locate a translocation breakpoint. This approach is able to establish whether or not a chromosomally balanced embryo carries the translocation [72], which is not possible by standard PGT-SR and PGT-A.

The same group also performed single-sperm haplotyping when no family members were available to phase haplotypes for PGT-M. They conducted target enrichment of MALBAC preamplified SNVs selected for haplotype analysis; on average, 74% of 60 SNVs were detectable in each single sperm cell, corresponding to 26% of ADOs. It was concluded that, with MALBAC-amplified, single-sperm DNA, SNV analysis can be achieved without target enrichment under 3× average genomic coverage [73].

### 6.6. Haploseek

Haploseek is a low-coverage, sequencing-based cPGT application exploiting PicoPLEX for embryo WGA. Although custom target design is not required, a prerequisite for the PGT cycle is pre-case SNP array assessment of the parents and affected child to generate whole genome haplotypes. Further SNP array data, together with sequencing data from the embryos, are integrated using a hidden Markov model, which predicts whether or not the parental-affected haplotypes have been inherited by each of the sequenced embryos across all chromosomes [74]. The required sensitivity for accurate haplotype and CNV prediction can be achieved at 0.3–1.4× coverage—an attribute that markedly reduces the cost per embryo. Furthermore, despite the low-coverage sequencing of single-cell samples, the proportion of SNVs in the MPS’s data passing quality control has been reported to range between 89.8 and 98.6% [74,75].

### 6.7. Universal cPGT

Recently, Chen et al. have developed a comprehensive WGS-based PGT tool capable of assessing monogenic disorders, aneuploidies, and chromosomal rearrangements without the requirement of additional family members and without the need of any pre-clinical work-up [76]. However, PGT-SR can only be performed if unbalanced translocation embryos are available as a reference to distinguish between balanced translocation carriers and normal embryos. Haplotyping for PGT-M is achieved by analysing already-retrieved embryos as a reference, rendering it impossible to analyse cases where direct mutation loci testing in embryos cannot be achieved by NGS (e.g., trinucleotide expansion disorders, intergenic deletions). In such cases, family haplotyping or additional targeted testing is necessary [76]. A minimum of 4× depth of embryo and 10× depth of parental DNA sequencing is sufficient for reliable PGT-M-direct genotyping (only SNVs/small indels can be assessed) and haplotyping. Exploiting MDA for embryo WGA, the ADO rate was reported by Chen et al. to reach 20% (lower than that of Karyomapping used for data validation), although only a modest number of embryos was evaluated [76].

## 7. Other WGA Opportunities

In 2018, Del Rey et al. attempted to perform cPGT using Illumina’s TruSight One sequencing panel. Unlike low genomic coverage WGA, the CNV profiles obtained using TruSight One data were widely scattered and highly discordant. The overall limited sensitivity of SNV and CNV calling rates may be associated with the targeted nature of the TruSight One panel, which was originally developed for bulk DNA analysis. The group also assessed the suitability of different WGA types for the given analysis. After sequencing, MDA covered almost 95% of the regions having at least 10× depth, followed by SurePlex and MALBAC, covering 65.5% and 64%, respectively [22].

### 7.1. Triplet Expansion Analysis

An expanded WGA application opportunity was demonstrated by Rajan-Babu and colleagues, who developed a direct mutation loci assessment PGT-M protocol for *FMR1* CGG trinucleotide expansion that causes Fragile X syndrome. PGT-M for Fragile X syndrome is currently performed by phasing maternal and paternal alleles directly or after WGA. A major limitation of this approach relates to its inability to amplify large GC-rich premutation and full-mutation *FMR1* alleles, making it difficult to discern homozygous normal female embryos from those heterozygous for a normal allele and a non-amplifiable premutation or full-mutation allele. Rajan-Babu et al.’s novel strategy utilizes MDA for WGA, followed by triplet-primed polymerase chain reaction (TP-PCR) for robust the detection of expanded *FMR1* alleles in parallel with linked multimarker analysis [77]. Notably, the authors have demonstrated that the premutation amplicon peak from single cells (direct PCR) is of significantly lower fluorescence intensity than the amplicon peaks from gDNA and WGA product. The same group has also developed a direct mutation loci assessment protocol for *DMPK* CTG expansion that causes myotonic dystrophy type 1. Testing prior to the development of this new protocol was problematic, as an unaffected embryo relied on the detection of the affected parent’s normal allele, which was only possible if it had a repeat size that was different from both alleles of the unaffected parent. The new strategy similarly utilizes MDA followed by the TP-PCR detection of expanded *DMPK* alleles [78]. The ADO rate of either the normal or expanded allele has been reported to be 22.4% for single-cell analysis, while multi-cell samples were amplified without ADO [78].

### 7.2. Single Gene Deletion Detection

Recently, Ren et al. demonstrated the successful direct detection of *DMD* gene deletions in preimplantation embryos. Their method of exploiting MALBAC amplification and 2× depth NGS without target enrichment can detect deletions of approximately 1 Mb [79].

### 7.3. Non-Invasive PGT

The feasibility of performing PGT using DNA obtained from blastocoel fluid or spent culture medium (termed non-invasive PGT) has been investigated. Despite non-invasive PGT initially appearing fairly promising, amplification failure occurred in an unacceptably high proportion of samples, reaching 81%. Clinically unsatisfactory results have been documented for SurePlex, MALBAC, as well as a modified MDA approach suitable for fragmented DNA [80,81,82,83,84], clearly demonstrating that the lack of success is not related to the WGA, but rather to the non-invasive nature of the approach. A comprehensive review of non-invasive PGT has recently been published by Leaver and Wells [85].

### 7.4. Long Read Sequencing

Although long read sequencing is currently used to detect translocation breakpoints for successful prospective PGT-SR [86,87], Peters et al. attempted to phase and detect de novo SNVs from blastocyst biopsies using Complete Genomics’ DNA nanoarray sequencing platform [88]. The protocol entails the apportionment of a single sample into a 384-well plate format, decreasing the fraction of the genome in each well to 10–20% of a haploid genome. DNA in each well is then amplified using a modified phi29-based MDA protocol to replicate each fragment ~10,000× [61,88]. Despite yielding 19–38× coverage of both the maternal and paternal alleles and mapping up to 97% of the heterozygous SNVs, a significant amount of SNVs cannot be easily mapped to individual alleles. Furthermore, genomes assembled from a small number of cells and requiring highly amplified DNA show a large number of false-positive SNVs arising from errors incorporated during amplification, sequencing, and mapping. Despite the application of several sophisticated bioinformatic algorithms and filters, several hundred to a thousand erroneously called de novo SNVs per embryo still prevail. At present, this approach is only a rudimentary model, and additional strategies are required to distinguish erroneous de novo events before its introduction into clinics. Specifically, utilizing it in its current format in clinical PGT could lead to the elimination of an excessive number of healthy embryos if these erroneous de novo variants were to be taken as putative [61,88]. Furthermore, high-depth MPS of the genome remains relatively expensive.

The main features of each PGT approach described above are listed in Table 1.

## 8. Concluding Remarks

To enable the genetic diagnosing of preimplantation embryos, all of the current cPGT solutions require clonal amplification of the template DNA. Consequently, WGA is more essential than ever before and has become one of the most important tools in the ever-developing field of human PGT [42]. The availability of substantial volumes of initially minute amounts of embryonic DNA generated by a single WGA round has made it possible to: (i) avoid embryo rebiopsy and repeat the analysis on account of assay failure or (long term) misdiagnosis or genotype mismatch of the birthed baby; (ii) shorten the turnaround time between referral and clinical cycle because the adaptation/validation of PCR reactions at the single-cell level can be omitted from the pre-clinical work-up [6]; and, most importantly, (iii) develop multifactor and comprehensive PGT.

MPS-based approaches are much more standardized and allow for high-throughput automation, reduced hands-on time, and minimization of the possibility of human errors—all at a reduced cost [6]. Hence, MPS-based approaches are regarded as the most powerful platforms for future PGT [89]. Several groups have already demonstrated the ability to perform mutation loci assessment with a shallow sequencing depth (2–4×) without the need for target enrichment [69,70,76]. Furthermore, the technical resolution of CNV detection has been demonstrated to be down to several kilobases [66]. While researchers compete to reduce the testing time and simplify the use of PGT methods, clinically, these objectives are not always justified and can result in painful and hard-to-correct errors. PGT has never been a simple method and, by its very nature, cannot be. Despite all the tempting emerging technical PGT opportunities, clinical PGT should continue to strictly adhere to the existing guidelines [6,56] and always bear in mind that patient safety is the number one priority. Orthogonal SNV validation should never be omitted unless there is convincing evidence to the contrary. The resolution of CNV detection should not be set unreasonably high to minimize the detection of artifacts resulting from WGA-introduced bias appearing as extensive biological heterogeneity, as this can potentially lead to normal embryos being discarded [59,66].

### 8.1. Best WGA Solution for Comprehensive PGT

Although numerous WGA techniques have been developed over the years, MDA, MALBAC, and hybrid WGA methodologies such as SurePlex are the ones most widely used in contemporary PGT. As the DNA originates from either a single cell or a few cells, all WGA technologies to a certain extent are prone to incomplete genomic coverage and amplification bias—drawbacks that hamper the SNV/CNV analysis of embryonic genomes. Each of the WGA types evaluated above has its own characteristics, advantages, and disadvantages to be considered. While it is a common assumption that PCR-based WGA methods are more suitable for CNV applications and MDA-based WGA systems are more appropriate for SNV analysis [60]—even the ESHRE PGT-M Working Group recommends MDA for PGT-M and PCR-based WGA as the method of choice for the detection of CNVs [6]—they all can be and are successfully exploited in clinical PGT, especially when using cPGT approaches [25,55].

Despite the fact that WGA reactions are very much influenced by an array of conditions (including hands-on time, reaction volume, starting amount of material, etc.), often in a stochastic manner, our review clearly demonstrates that MDA, MALBAC, and hybrid methods are reliable, clinically valid, and time-tested WGA methods suitable for multilateral use. PicoPLEX and its analogues were initially adapted for PGT and are now the foremost solution for PGT-A using arrays and NGS. Unlike the other WGA techniques, MALBAC comes from the MPS era, and its applications are mostly associated with MPS. MDA appears to be slightly more popular than the other WGA types when it comes to cPGT solutions exploiting MPS (Table 1). The advantages of MDA are associated with its utilization of the proofreading enzyme phi29, allowing for strand displacement DNA synthesis without dissociating from the template for more than thousands of nucleotides under isothermal conditions. Nonetheless, currently no one method suits all the criteria. However, when opting for cPGT, well-designed bioinformatic tools that unambiguously detect all types of genetic changes may provide a compromise [90].

While there are numerous studies comparing the performance of different WGA techniques, there is a limited number of these studies relating to PGT and even fewer in relation to human embryonic cells. Therefore, studies conducting an unbiased comparison of different WGA types that consistently and systematically follow certain conditions with regard to MPS parameters and same-stage embryo biopsies would be very welcome additions to the literature. New unbiased studies might highlight additional aspects of WGA performance in cPGT not determinable from the assessment of other cell types. As we have discussed above, different MPS parameters can greatly influence the semblance of WGA performance (e.g., genomic coverage depends on sequencing depth).

### 8.2. Future Perspectives

As WGA techniques are continually being improved, the majority of drawbacks of the current techniques should be minimized in the near future. In addition, new solutions are expected to be developed. For example, a few years ago, the newly cloned and characterized *Thermus thermophilus* DNA primase—*Tth*PrimPol—was integrated into the phi29-based MDA protocol. In this system, termed TruePrime, *Tth*PrimPol primase randomly initiates DNA amplification by synthesizing new primers on the native DNA and displaced strands, removing the need for random primer exploitation [91]. It was predicted that this random primer-free WGA would display minimal amplification bias compared to methods exploiting random primers. However, although the trial sequencing run of TruePrime samples was of high quality, with the reads almost exclusively mapping to human genomic material, after deduplication, only 22% of the reads remained, highlighting the magnitude of the problem of duplicated reads [92]. Nonetheless, refinement of this exciting technology is eagerly awaited.

Another WGA improvement might come from the usage of droplet or chambered microfluidic technologies. While advances in microfluidics have been predicted to enter IVF laboratories [93], their usage in human PGT, to the best of our knowledge, has not been demonstrated. In microfluidic devices, molecules are isolated in compartments. Therefore, they amplify to saturation without competing for resources. This improves genomic coverage and reduces amplification bias, thus enhancing the overall quality of sequence data [94]. To date, this has been demonstrated for MDA [95,96] and with improved reproducibility for MALBAC [97]. The usage of microfluidics in comparison to conventional in-tube reactions shortens the turnaround time, reduces the likelihood of contamination, and decreases reaction expenses, as the reaction is performed in a smaller volume [98]. We anticipate that microfluidic usage also has the potential to improve PGT applications.

On a closing note, we can conclude that the future of PGT with comprehensive solutions is already here. By following the best practice guidelines for PGT and a patient-tailored approach, in combination with state-of-the-art instrumentation, at the heart of which lies WGS, it is now possible to minimize the number of technical errors and provide the maximum number of confident clinical diagnoses. These outcomes will only seek to strengthen and extend the practice of PGT.

## Figures and Tables

**Figure 1 ijms-23-04819-f001:**
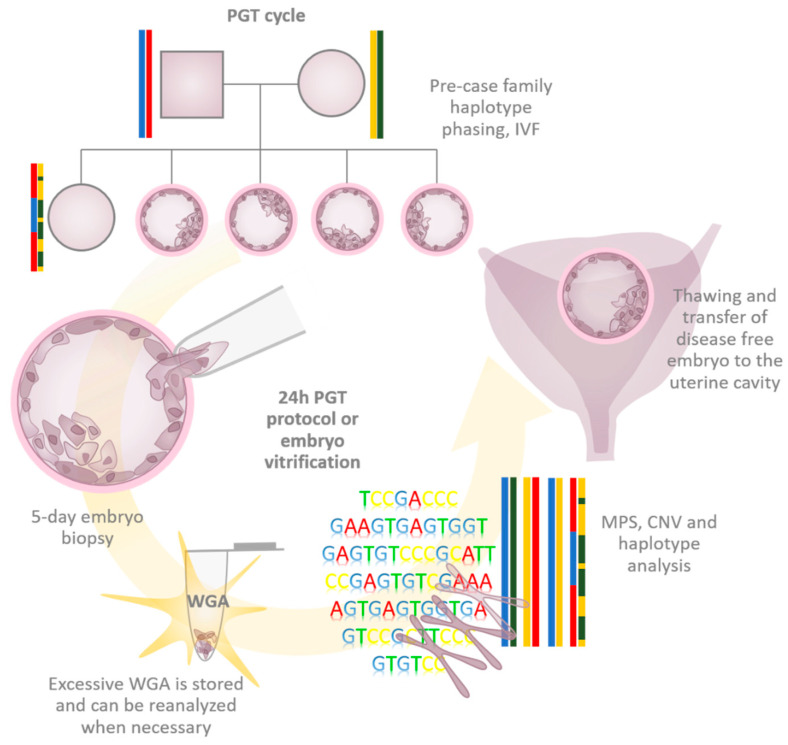
Clinical cycle of comprehensive preimplantation genetic testing. The process starts with family genetic testing in order to establish normal and disease-associated family haplotypes. Next, oocyte stimulation, retrieval, and fertilization are performed. Embryos surviving until day 5 undergo laser-assisted hatching and biopsy. Biopsy material consisting of a few trophectoderm cells is amplified using one of the WGA techniques. WGA material is further used to assess the embryonic genome, including haplotype, causative variant, aneuploidy, and CNV analyses. The disease-free embryo is subjected to embryo transfer to the uterine cavity. The whole cycle can be repeated within 24 h, with a day-6 embryo being transferred. If the genetic testing takes longer, then embryos are vitrified and thawed before being transferred. PGT—preimplantation genetic testing, WGA—whole-genome amplification, IVF—in vitro fertilization, MPS—massively parallel sequencing, CNV—copy number variation.

**Figure 2 ijms-23-04819-f002:**
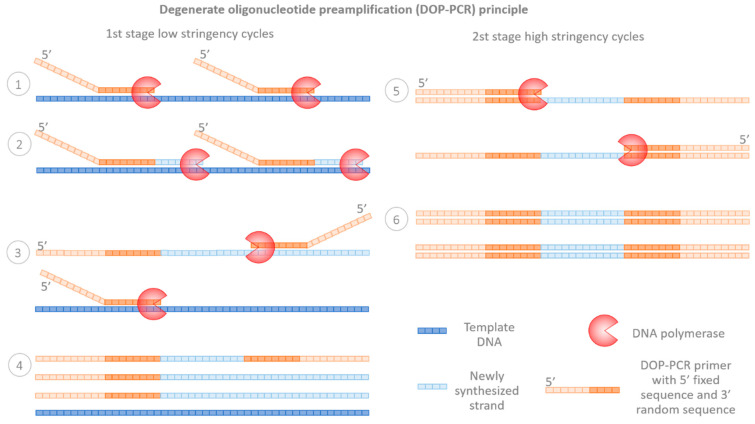
Principle of degenerate oligonucleotide primer (DOP) PCR. (**1**–**4**) During the first stage, DOP-PCR primers non-specifically bind to and amplify template DNA under low-stringency conditions. (**5**,**6**) During the second stage, primers target the stage one-synthesized amplicons under high-stringency conditions, producing copies of the copies.

**Figure 3 ijms-23-04819-f003:**
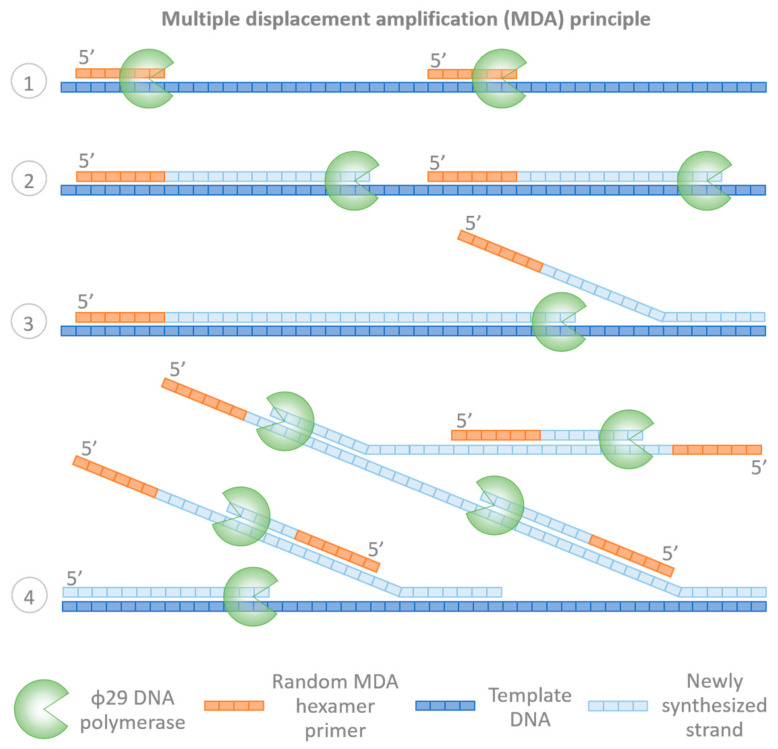
Principle of multiple displacement isothermal amplification (MDA): (**1**) During the annealing step, random MDA hexamer primers bind to the template DNA. (**2**,**3**) Strand elongation is achieved using phi29 DNA polymerase with strand displacement activity. (**4**) The reaction produces a hyperbranched DNA structure.

**Figure 4 ijms-23-04819-f004:**
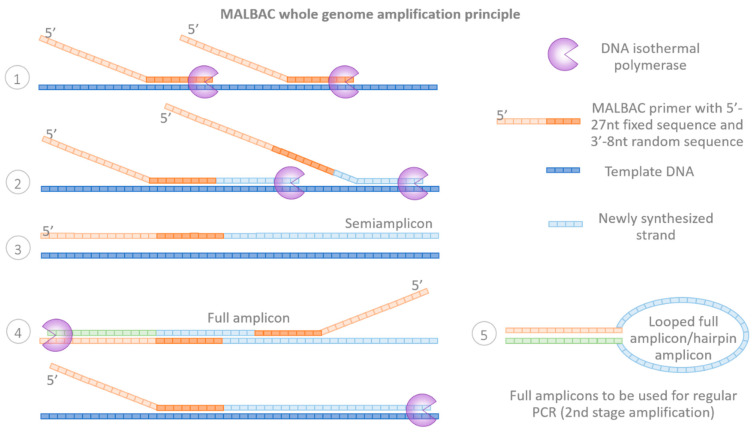
Principle of MALBAC whole-genome amplification: (**1**) MALBAC first stage preamplification is initiated with the annealing of partially random primers. (**2**) Strand elongation is achieved using a DNA polymerase with strand displacement activity. (**3**,**4**) Amplification of the semi-amplicons produces full amplicons with complementary 3′ and 5′ ends. (**5**) Looping prevents the full amplicons from further amplification during the first stage PCR and they instead serve as the target during the second stage regular PCR.

**Figure 5 ijms-23-04819-f005:**
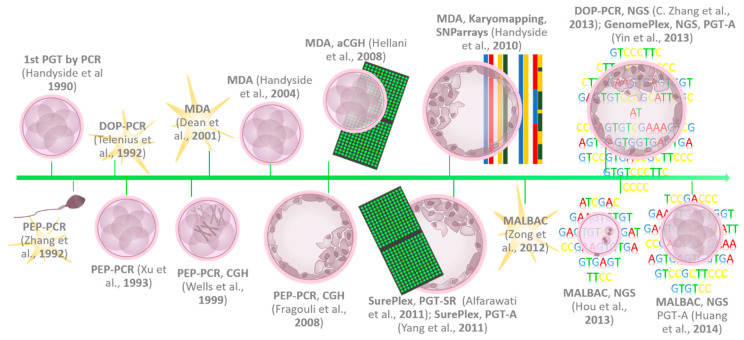
Timeline of the key developments in WGA and milestones of PGT. PGT-A/SR—preimplantation genetic testing for aneuploidies/structural rearrangements; WGA—whole-genome amplification; DOP-PCR—degenerate oligonucleotide primer PCR WGA technique; PEP-PCR—primer extension preamplification PCR WGA technique; MDA—multiple displacement amplification WGA technique; aCGH—array comparative genomic hybridization; MALBAC—multiple annealing and looping-based amplification cycles WGA technique; SNP—single nucleotide polymorphism; NGS—next generation sequencing [2,3,15,20,21,23,26,28,31,32,36,37,38,39,40,41].

**Table 1 ijms-23-04819-t001:** Comprehensive PGT solutions exploiting MPS.

	Clinically Validated	Principle	WGA	Custom Target Design Needed	CNV	Whole-Genome Haplotyping	Direct Mutation Loci Analysis	Balanced Translocation Carrier Embryos	24-h Protocol	Pre-Clinical Work	No-Cost Multiplexing	Citation
Karyomapping	Yes	SNP array/MPS	MDA	No	Yes	Yes	No	Yes	Yes	Yes, family haplotyping	No	[28,67]
Haplarithmisis	Yes	SNP array/MPS	MDA	No	Yes	Yes	No	Yes	Yes	Yes, family haplotyping	No	[62]
Haploseek	Yes	Low coverage WGS	PicoPLEX	No	Yes	Yes	No	Yes	Yes	Yes, family haplotyping using SNP arrays	Yes	[74]
Universal cPGT by Chen et al.	Yes	Low coverage WGS	MDA	No	Yes	Yes	Yes (target sequencing)	Yes, if unbalanced embryo is available	No	Haplotyping is performed after embryo MPS data are available, using mutated or unbalanced embryos as a reference	Yes	[76]
OnePGT	Yes	Reduced representation sequencing	MDA	No	Yes	Yes	No	Yes	Unknown	Yes, family haplotyping	Yes	[68]
MARSALA	No	Low coverage WGS	MALBAC	Yes	Yes	No	Yes (target sequencing)	No	Yes	Yes, family haplotyping	Yes	[69,70]
MaReCs	No	High coverage high resolution WGS	MALBAC	Yes	Yes	No	No	Yes	No	Haplotyping is performed after embryo CNV assessment	Yes	[72]
Triplet expansion analysis	No	Triplet-primed PCR	MDA	No	NA	NA	Yes	NA	Yes	NA	No	[77,78]
Single gene deletion detection	No	Low coverage WGS	MALBAC	No	Yes	Yes	Yes	NA	Unknown	Yes	Yes	[79]
Long-read technology	No	High coverage long-read sequencing	MDA	No	NA	Yes	Theoretically yes	NA	No	No	No	[88]

## Data Availability

All data are incorporated into the article and its online Appendix A.

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
