# Peer review of "Whole Genome Amplification in Preimplantation Genetic Testing in the Era of Massively Parallel Sequencing"

_ijms, 2022, doi:10.3390/ijms23094819_

Round 1
Reviewer 1 Report
Minor comments:
- What is the prevalence of PGS in the clinical setting?
- Do PGS procedures negatively impact the developing human embryo and if so, how, especially in later developmental stages?
- The rate of aneuploidy in IVF has been shown to be influenced by the type of ovarian stimulation used (mild vs. conventional) and the inclusion of luteinizing hormone (LH) supplementation.
What is the author’s opinion?
- Strategy, technology and techniques for performing PGS vary between IVF centers. Can a standard procedure be made?
- Day-5 embryo biopsy or day-3 embryo biopsy. Difference and benefits?
- Kindly comment on Ethical considerations pertaining to PGS especially highlighting the Middle East population.
- Can you comment on NIPT as well?
- PMID: 33613643, PMID: 33804821, PMID: 35300641, PMID: 34937515
Reviewer 2 Report
Congratulations on this comprehensive review of the different PGT techniques. You give a broad overview, your illustrations are very fitting and especially the timeline is very descriptive. I would only recommend to somewhat shorten the future perspectives section.
Author Response
Dear Reviewer,
we would like to thank you for the quick and positive review of our work. We have now deleted a couple of lines within the Future Perspectives section. The changes to the text are marked using the Track changes option.
Kind regards,
Ludmila